# Myo-inositol versus D-chiro-inositol in murine in vitro follicular development: An experimental study relevant to polycystic ovary syndrome

Aika Korai[1], Tsuyoshi Baba🆔[1]*, Fukiko Kasuga-Yamashita[1], Sachiko Nagao[1], Yuya Fujibe[1], Miyuki Morishita[1], Yoshika Kuno[1], Tasuku Mariya🆔[1], Keiko Ikeda[2], Hiroyuki Honnma[2], Toshiaki Endo[1], Tsuyoshi Saito[1]

1 Department of Obstetrics and Gynecology, Sapporo Medical University, Sapporo, Hokkaido, Japan,
2 Sapporo ART Clinic, Sapporo, Hokkaido, Japan

* tbaba@sapmed.ac.jp

## Abstract

Inositol plays a crucial role in follicular development by regulating insulin signaling and ovarian function. However, its precise mechanism of action remains unclear. This study investigated the effects of myo-inositol (MI) and D-chiro-inositol (DCI) on the development of murine ovarian follicles in vitro. Follicles treated with DCI exhibited larger diameters than controls on Day 6 (275.20 ± 12.54 μm; p = 0.037) and Day 8 (277.47 ± 11.47 μm; p = 0.048), indicating a modest, marginally significant effect that was not maintained by Day 10. The rate of follicular antrum formation was significantly higher in the DCI-treated group on Day 6 (p < 0.05); however, no significant differences were observed on Days 8 and 10. In contrast, MI treatment did not affect follicular survival, diameter, or antrum formation compared with controls. Estradiol concentrations and the expression levels of follicle-stimulating hormone receptor and aromatase genes did not differ significantly among groups. Together, these data provide in vitro evidence that DCI can facilitate the transition from the secondary (preantral) to the early antral stage under these culture conditions. Given the small experimental sample size, the use of healthy murine follicles cultured under a high FSH concentration, and the absence of a PCOS-like ovarian milieu, these findings should be interpreted cautiously and cannot be directly generalized to infertility treatment in women with PCOS. Future studies using PCOS animal models and human follicle systems are needed to clarify translational relevance of these findings.

## Introduction

Polycystic ovary syndrome (PCOS) is a common endocrine disorder characterized by ovulatory dysfunction, hyperandrogenemia, and polycystic ovarian morphology that affects 5–20% of women of reproductive age [1]. A significant proportion of patients diagnosed with PCOS manifest symptoms of insulin resistance and hyperinsulinemia

**Data availability statement:** All relevant data are within the manuscript.

**Funding:** The author(s) received no specific funding for this work.

**Competing interests:** The authors have declared that no competing interests exist.

[2]. In addition to these metabolic features, ovulatory dysfunction in PCOS is driven by tightly coupled neuroendocrine and intraovarian abnormalities, which together create a "follicle excess but follicle arrest" phenotype [3–5].

The mechanism of ovulatory failure in PCOS is considered twofold: accelerated development up to the early antral follicle stage, and arrested development thereafter. Elevated luteinizing hormone (LH) levels induce androgen production in theca cells and androgen accelerates development from secondary follicles to small antral follicles. However, a high-LH environment inhibits follicle-stimulating hormone (FSH)-dependent follicle development by suppressing FSH receptor expression [6]. Even if LH stimulation causes secondary follicles to grow into antral follicles, these antral follicles may exhibit reduced responsiveness to FSH. As a result, a large number of small antral follicles accumulate in the ovary but do not ovulate, resulting in polycystic ovarian morphology [6]. Mechanistically, persistently rapid GnRH pulsatility in PCOS has been suggested to favor LH hypersecretion and an elevated LH/FSH ratio, potentially creating an LH-dominant endocrine milieu in which the FSH-dependent granulosa cell differentiation required for dominant follicle selection and preovulatory maturation may be less likely to proceed normally [3–5]. In addition, anti-Müllerian hormone (AMH), often elevated in PCOS due to an increased pool of small antral follicles, has been proposed to attenuate follicular sensitivity to FSH and to limit aromatase induction, which could contribute to impaired follicle selection and maturation [7,8]. Taken together, ovulatory failure in PCOS may be viewed as an expansion of the small antral follicle cohort accompanied by reduced FSH responsiveness and suboptimal selection/maturation, culminating in anovulation and polycystic ovarian morphology [3,7,8].

The pathophysiology of PCOS is complex and not fully understood; however, insulin resistance is believed to be a major contributing factor. Insulin resistance affects the insulin signaling pathway during ovarian steroidogenesis, leading to abnormal androgen production [4]. Therefore, in this study, we focused on inositol, a compound involved in insulin resistance and considered a factor in follicle development [9]. Inositol exists as nine isomers, but myo-inositol (MI) and D-chiroinositol (DCI) are particularly abundant in the follicular fluid and play important roles in insulin signaling and follicle development in the ovary.

MI originates from the diet and functions as a precursor to inositol triphosphate, a second messenger that regulates hormones such as FSH and insulin [10]. DCI is synthesized by an epimerase that converts MI into DCI, which functions as an aromatase inhibitor in the ovaries [11]. The balance between the two isomers supports normal hormone production and ovarian function. The MI/DCI ratio in follicular fluid is 100:1 in healthy individuals, it decreases to 0.2:1 in insulin-resistant PCOS patients [12]. In women with PCOS, insulin resistance causes decreased epimerase activity in normal tissues, leading to systemic DCI deficiency. Conversely, insulin sensitivity is maintained in the ovary, where hyperinsulinemia increases epimerase activity, causing a pathological increase in DCI within the ovary. This phenomenon is known as the ovarian paradox [13]. This disruption in the MI/DCI ratio is thought to be involved in the pathogenesis of PCOS.

Clinical data on inositol supplementation in PCOS are mixed, with heterogeneity in formulations, dosing, and outcomes, and the overall certainty of evidence for reproductive endpoints remains limited. Recent international evidence-based PCOS guidelines emphasize that metformin has greater efficacy for metabolic features than inositol and that the clinical benefits of inositol are limited; therefore, inositol is not considered an evidence-based, first-line therapy for infertility in PCOS and, at most, may be considered via shared decision-making in selected individuals [14,15]. Nevertheless, both MI and DCI have been investigated for potential effects on metabolic and hormonal parameters in PCOS, and combination regimens (e.g., MI:DCI 40:1) have been proposed, although definitive evidence for improved fertility outcomes remains lacking.

Despite extensive clinical and metabolic research, a key knowledge gap remains: there are few controlled in vitro studies that isolate the direct, stage-specific actions of MI versus DCI on folliculogenesis independent of systemic endocrine and metabolic factors. In particular, it is unclear whether changes in MI/DCI exposure can directly influence the secondary-to-early antral transition, a stage implicated in the early acceleration observed in PCOS. To address this gap, we examined the effects of MI and DCI on isolated murine secondary follicles using a single-follicle culture platform, enabling quantification of survival, growth, antrum formation, steroid output, and selected granulosa-cell transcripts under defined conditions [16–18].

## Materials and methods

### Ethics approval and consent to participate

Animal procedures were carried out in compliance with the regulations of Sapporo Medical University and the Scientists' Center for Animal Welfare. The experimental protocol was reviewed and authorized by the Institutional Animal Care and Use Committee (approval No. 22–051). Consent to participate was not required for this study.

### Animal and Cell Culture

Female ICR mice were purchased from Sankyo Laboratory Services (Sapporo, Japan). The animals were euthanized prior to tissue collection. Euthanasia and subsequent handling were performed in accordance with the guidelines of Sapporo Medical University and the Center for Animal Protection Scientists. The study protocol was approved by the Institutional Animal Care and Use Committee (approval number: 22–051). Six-week-old female ICR mice (n = 16) were euthanized via intraperitoneal administration of pentobarbital (120 mg/kg). The ovaries were excised, and secondary follicles measuring 130–160 µm in diameter were isolated mechanically with a 30-gauge needle under an inverted microscope. Follicles were chosen only when they exhibited an intact basement membrane, well-defined granulosa cell layers, and centrally located spherical oocytes. To exclude factors that control follicle growth and steroidogenesis, such as pituitary gonadotropins, steroid hormones, and local growth factors, each follicle was placed individually in 48-well multicellular repellent surface plates (Greiner Bio-One International GmbH, Kremsmünster, Austria). Each well contained 5% fetal bovine serum (Corning, Corning, NY, USA), 6 µg/mL insulin, 5.5 µg/mL transferrin, 6.7 ng/mL sodium selenite, 200 IU/mL penicillin (Thermo Fisher Scientific K.K., Tokyo, Japan), 66 mIU/mL FSH (Sigma-Aldrich Japan, Tokyo, Japan). Earlier reports indicated that the threshold FSH concentration needed to achieve the maximal growth response was 67 mIU/mL [19]. Follicles were maintained at 37 °C in a humidified incubator with 5% $CO_2$. The culture medium was refreshed every other day by replacing half of the volume with fresh medium, and culture was continued for a total of 10 days.

To investigate the influence of DCI and MI on early follicular development, these compounds (Sigma-Aldrich, Japan) were supplemented into the culture medium. Follicles collected from eight mice (12 follicles per mouse per group) were allocated at random to one of three conditions: (1) control (CTRL), consisting of the base medium with the vehicle for DCI (dimethyl sulfoxide; DMSO); (2) DCI 20, CTRL medium containing 20 µM DCI and DMSO; or (3) DCI 50, CTRL medium containing 50 µM DCI and DMSO. In four mice, follicle survival and growth were measured and E2 levels were

determined in the follicular culture supernatant obtained from their ovaries. In the remaining four mice, RNA extraction, reverse transcription, and real-time quantitative polymerase chain reaction were performed on follicular cultures obtained from their ovaries. Follicles obtained from another eight mice (12 follicles per mouse per group) were similarly distributed into three groups: (1) control (CTRL), base medium with the MI vehicle (DMSO); (2) MI 20, CTRL medium supplemented with 20 μM MI and DMSO; or (3) MI 50, CTRL medium supplemented with 50 μM MI and DMSO. The DCI and MI doses used in this study were based on the results reported by Unfer, et al. [20]. In the same report, MI concentrations in the oocyte fluid were $3.00 \pm 1.22$ μM for PCOS patients and $22.68 \pm 1.38$ μM for healthy women; $p < 0.001$). The DCI concentration in the follicular fluid was $16.65 \pm 2.01$ μM for PCOS patients and $0.22 \pm 0.08$ μM for healthy women; $p < 0.001$.

### Follicle survival and growth

Follicle survival and growth were assessed on Days 1, 6, 8, and 10 using an SMZ18 inverted microscope (Nikon, Tokyo, Japan). Follicles were considered degenerate when the oocytes turned black or were expelled from the follicle, when the granulosa cells appeared degenerated or darkened, or if the follicle diameter showed a reduction. Follicular diameter was calculated as the mean of two perpendicular measurements, obtained with the NIS-Elements Documentation software (version D 3.22.00; Nikon).

### Measurement of E2

Estradiol (E2) concentrations in the culture medium were measured on Days 5 and 9 using an ELISA kit (R&D Systems, Minneapolis, MN, USA; detection range: ~0–3.0 ng/mL). E2 concentrations were measured individually from the culture medium collected from each well. The inter- and intra-assay coefficients of variation for the kits were less than 10%.

### RNA extraction, reverse transcription, real-time quantitative polymerase chain reaction

On the seventh day of culture, 6–10 follicles from each experimental group were selected for mRNA analysis. Individual follicles were disrupted using a 30-gauge needle, and the follicle walls along with cumulus cells were collected for RNA extraction. Total RNA was purified following the manufacturer's protocol using the Absolute RNA Nanoprep Kit (Agilent, Santa Clara, CA, USA). Complementary DNA (cDNA) was then synthesized from 1 μg of total RNA with the Superscript II Reverse Transcriptase kit (Thermo Fisher Scientific). Quantitative PCR was conducted using TaqMan gene expression assays on the AB StepOnePlus Real-Time PCR System (Thermo Fisher Scientific K.K.). The following assays were employed: *Fshr* (Assay ID: Mm00442819_m1), aromatase (*Cyp19a1*, Assay ID: Mm00484049_m1), with 18S ribosomal RNA (Assay ID: Mm03928990_g1) serving as the internal control. The PCR protocol included 40 cycles of denaturation at 95°C for 15 seconds followed by annealing/extension at 60°C for 60 seconds. Relative gene expression levels were calculated using the 2^-ΔΔCt method.

### Sample size and power

To adhere to the 3Rs principle, particularly the principle of reduction, a single follicle culture system was adopted. This allowed multiple experimental units to be obtained from each animal. In total, follicles were collected from 16 female ICR mice. To avoid allocation bias, follicles from both ovaries were used, with 12 follicles randomly assigned to each group.

This experimental design aligns with established single follicle culture studies (our previous work [20]), where 3–5 animals per experiment were sufficient to detect biologically meaningful differences.

Considering the observed range of follicle diameter variation (standard deviation approximately 10–15 μm), this sample size was deemed adequate to detect a moderate effect using one-way ANOVA ($\alpha = 0.05$).

## Data analysis

Data are expressed as the mean±standard error of the mean (SEM). Differences between treatment groups were analyzed using one-way analysis of variance (ANOVA) followed by Student–Newman–Keuls post hoc test, performed with SigmaPlot version 13.0 (Systat Software, San Jose, CA, USA). A p-value less than 0.05 (two-tailed) was considered statistically significant.

## Results

### Effects of DCI and MI on secondary follicle development in mice

The follicle survival rate in the DCI experiment ranged from 76% to 88%, with no significant differences between the groups (p=0.161) (Fig 1a). In terms of follicle diameter, the DCI 20 μM group (275.20±12.54 μm; p=0.037) and the DCI 50 μM group (277.47±11.47 μm; p=0.048) were significantly larger than those in the CTRL group (236.23±13.34 μm) on Day 6. In addition, on Day 8, the follicle diameter in the DCI 20 μM group (318.96±10.43 μm; p=0.012) and the DCI 50 μM group (332.05±12.59 μm; p=0.002) were significantly greater than that of the CTRL group (274.16±11.64 μm). However, by Day 10, no significant differences were observed between groups (Fig 2a). In terms of the antrum formation rate, the DCI 20 μM group (65.3±6.55%; p=0.037) and the DCI 50 μM group (70.8±14.2%; p=0.046) had significantly greater rates than that of the CTRL group (31±6.39%) on Day 6, but on Days 8 and 10 the differences were no longer statistically significant (Fig 3a). However, since the p-value was close to 0.05, the increase in follicle diameter after DCI treatment should be considered a modest effect rather than a robust one.

The follicle survival rate in the MI experiment ranged from 83% to 88%, with no significant differences between groups (p=0.700) (Fig 1b). No significant differences were observed in follicle diameter among the groups (Fig 2b). The antrum formation rates in the CTRL, MI 20 μM, and MI 50 μM treatment groups were 41.1%, 56.2%, and 55.2%, respectively (p=0.780), showing no significant differences (Fig 3b).

### Effect of DCI and MI on Estradiol (E2) production

The E2 concentration in the culture medium on Day 5 of the DCI experiment in CTRL, DCI 20 μM, and DCI 50 μM groups was 0.45±0.03 ng/mL, 0.53±0.03 ng/mL, and 0.60±0.06 ng/mL, respectively (p=0.108) (Fig 4a). On Day 9, the E2

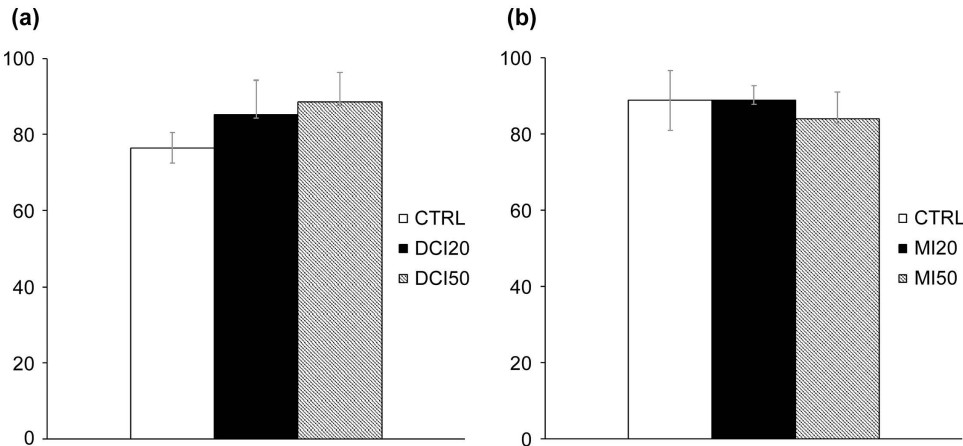

**Fig 1. Effects of inositol on follicle survival rates.** Abbreviations: CTRL (control), DCI (D-chiro-inositol), MI (myo-inositol), DMSO (dimethyl sulfoxide). **(a)** Survival on Day 10 for CTRL, DCI 20 μM, and DCI 50 μM. The CTRL group received base medium with vehicle (DMSO); DCI groups received base medium with 20 or 50 μM DCI plus DMSO. **(b)** Survival on Day 10 for CTRL, MI 20 μM, and MI 50 μM (base medium with MI 20 or 50 μM plus DMSO). Data are presented as mean±standard error. One-way ANOVA: no significant differences (panel a: p=0.161; panel b: p=0.700).

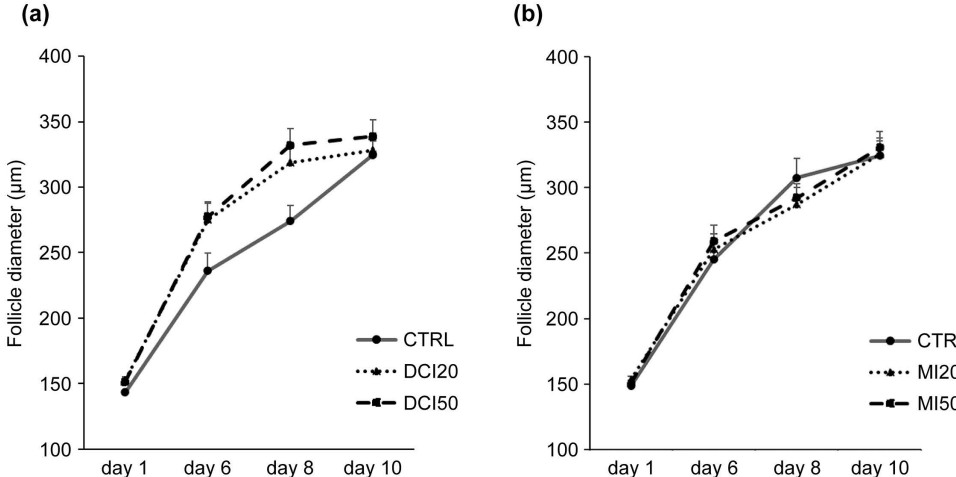

**Fig 2. Effects of inositol on average follicle diameters.** Abbreviations: CTRL, DCI, MI, DMSO. **(a)** DCI; **(b)** MI. Follicle growth was tracked at Days 6, 8, and 10; treatments as in Fig 1. Data are mean±standard error. *One-way ANOVA with Student–Newman–Keuls post hoc; $p < 0.05$ considered significant.

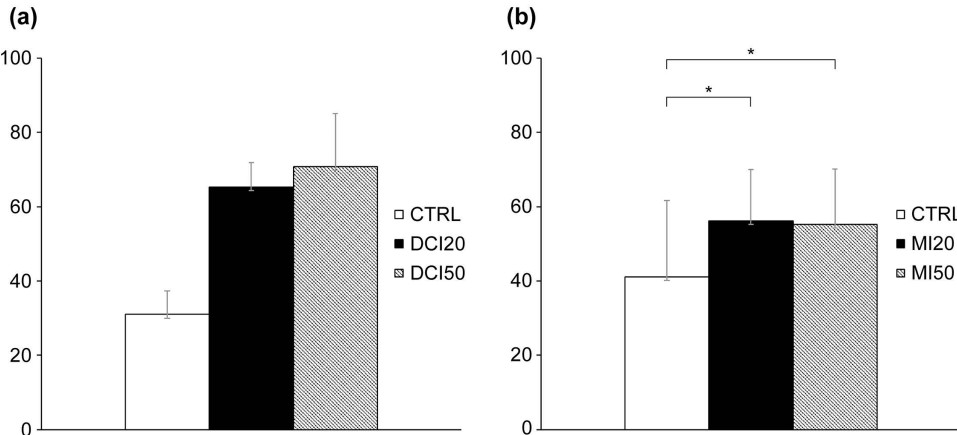

**Fig 3. Effects of inositol on antrum formation rate on Day 6.** Abbreviations: CTRL, DCI, MI, DMSO. **(a)** DCI; **(b)** MI. Assessments at the indicated time points; treatments as in Fig 1. Data are mean±standard error. *One-way ANOVA with Student–Newman–Keuls post hoc; $p < 0.05$ considered significant.

concentration in the CTRL, DCI 20 μM, and DCI 50 μM groups was 0.69±0.08 ng/mL, 0.75±0.10 ng/mL, and 0.85±0.07 ng/mL, respectively ($p = 0.444$) (Fig 4b), with no statistically significant differences in the E2 concentrations among the groups.

For MI, the E2 concentration in the culture medium on Day 5 in the CTRL, MI 20 μM, and MI 50 μM groups was 0.41±0.06 ng/mL, 0.46±0.09 ng/mL, and 0.39±0.09 ng/mL, respectively ($p = 0.845$) (Fig 4c). The E2 concentration on Day 9 in the CTRL, MI 20 μM, and MI 50 μM groups was 1.36±0.3 ng/mL, 1.04±0.19 ng/mL, and 0.84±0.11 ng/mL, respectively ($p = 0.256$) (Fig 4d). No statistically significant differences in E2 concentrations were observed between groups.

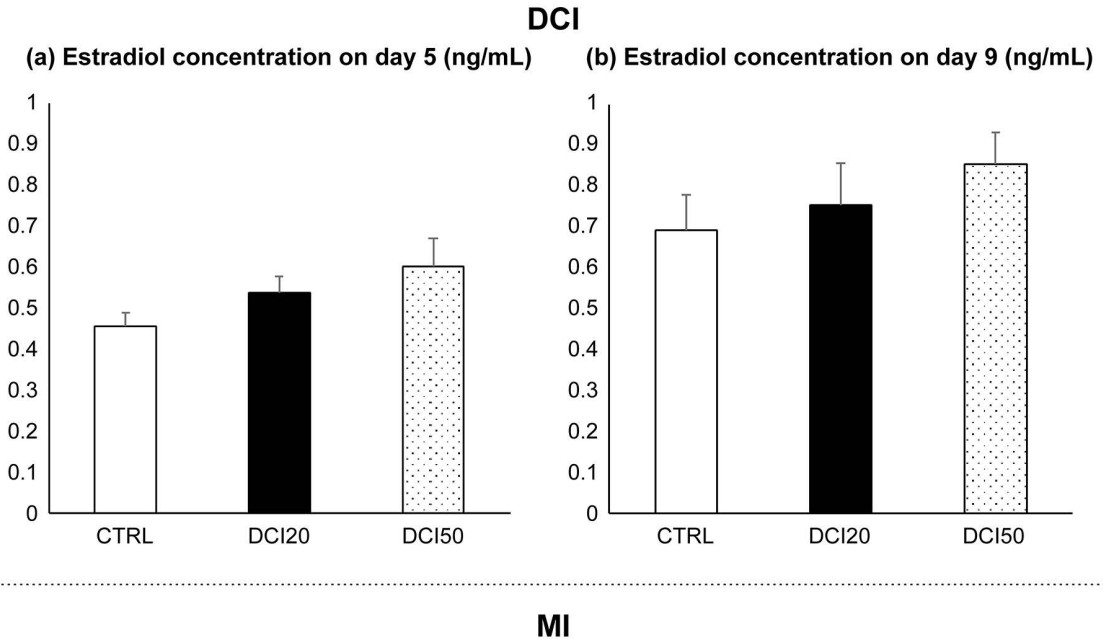

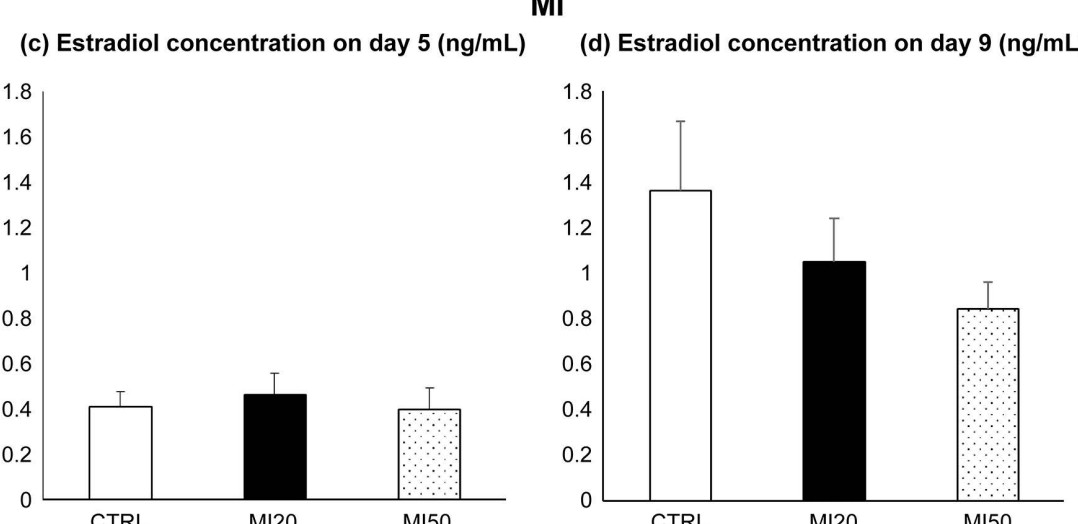

**Fig 4. Estradiol (E2) concentrations in culture media.** Abbreviations: E2, CTRL, DCI, MI, DMSO. **(a)** DCI Day 5; **(b)** DCI Day 9; **(c)** MI Day 5; **(d)** MI Day 9. Follicles were treated as in Fig 1. Data are mean ± standard error. *One-way ANOVA with Student–Newman–Keuls post hoc; p < 0.05 considered significant.

### The effects of DCI and MI on mRNA expression of the FSH receptor and aromatase gene

Relative mRNA levels of the FSH receptor (*Fshr*) and aromatase (*Cyp19a1*) in the DCI and MI experimental groups are shown in Fig 5. In the experimental group treated with DCI, levels of *Fshr* mRNA did not differ significantly between groups (p = 0.724) (Fig 5a), and no statistically significant differences were observed in *Cyp19a1* mRNA levels (p = 0.651) (Fig 5b). In the MI experiment, *Fshr* mRNA levels were not significantly different between the groups (p = 0.494) (Fig 5c). mRNA levels of *Cyp19a1* were lower in the MI group than in the CTRL group; however, this difference was not significant (p = 0.531) (Fig 5d).

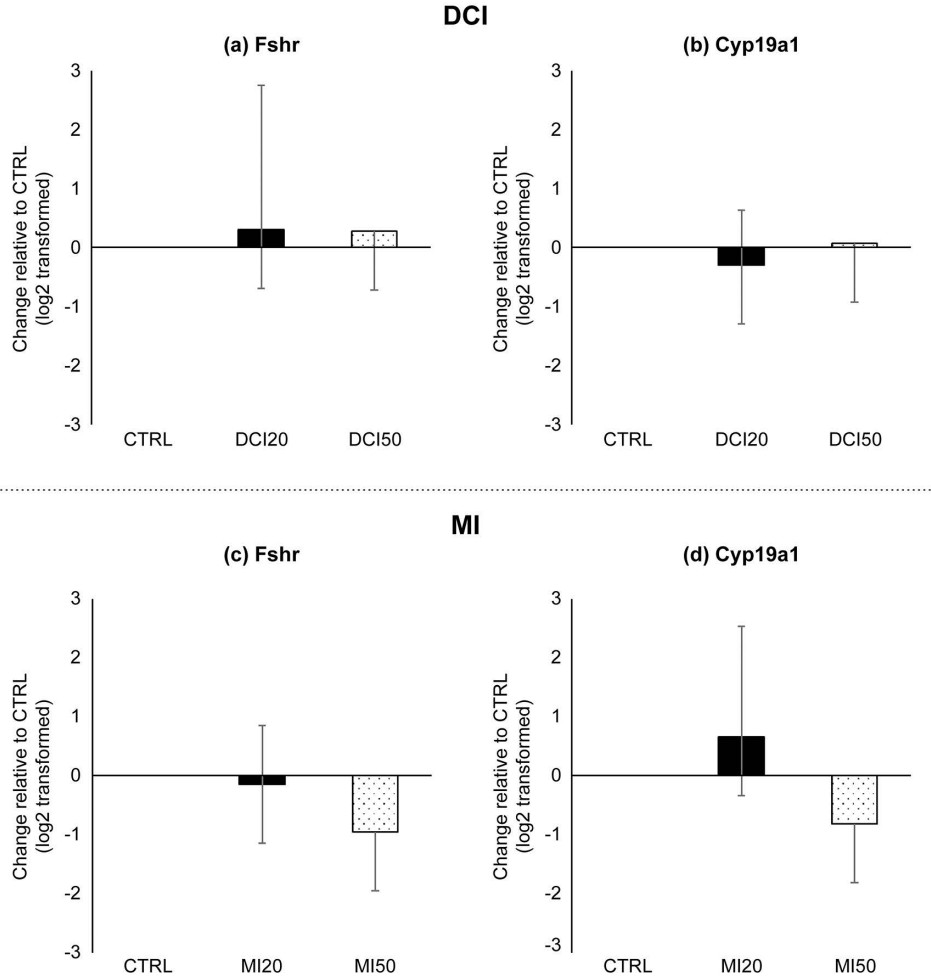

**Fig 5. Effects of inositol on *Fshr* and *Cyp19a1* expression in isolated follicles.** Abbreviations: *Fshr*, *Cyp19a1*, CTRL, DCI, MI, DMSO. Follicles were cultured for 10 days before mRNA analysis. Data are presented as log2(fold-change vs CTRL). DCI group: **(a)** *Fshr*; **(b)** *Cyp19a1*. CTRL=vehicle (DMSO); DCI 20=20 μM DCI+DMSO; DCI 50=50 μM DCI+DMSO. MI group: **(c)** *Fshr*; **(d)** *Cyp19a1*. CTRL=vehicle (DMSO); MI 20=20 μM MI+DMSO; MI 50=50 μM MI+DMSO. Data are mean±standard error. *One-way ANOVA with Student–Newman–Keuls post hoc; p<0.05 considered significant.

## Discussion

In this study, DCI supplementation in a single-follicle murine culture system was associated with a modestly earlier increase in follicle diameter and a higher antrum formation rate on Day 6, suggesting facilitation of the secondary-to-early antral transition. However, the diameter differences were small, only marginally significant at some time points (p-values near 0.05), and not sustained through Day10.

Follicles cultured without DCI initially showed delayed development but ultimately reached the same stage of development as follicles exposed to DCI. Therefore, DCI supplementation in our system did not reproduce the late-stage arrest typical of PCOS. This indicates that the observed effect is stage-limited and context-dependent. Because our experiments used healthy murine follicles and did not recreate the hyperandrogenic, hyperinsulinemic, or inflammatory ovarian milieu characteristic of PCOS, we cannot infer that a lower MI/DCI ratio is causally responsible for PCOS follicular

dynamics. Accordingly, these findings should be interpreted as mechanistic, stage-specific in vitro observations rather than clinical conclusions. Future studies should directly test MI and DCI in established animal models of PCOS (e.g., androgen- or letrozole-induced models) and, where feasible, in human follicle culture systems to evaluate whether similar stage-dependent effects occur in a PCOS-relevant context.

The molecular mechanism by which DCI promotes follicular development remains unclear. Previous studies have shown that the inhibition of aromatase activity by letrozole reduces the conversion of testosterone to estradiol and significantly increases FSHr expression [20]. Based on these findings, we anticipated that DCI, which also suppresses aromatase activity, would decrease E2 production and enhance FSHr expression. However, in the present study, DCI treatment did not alter E2 levels, aromatase, or FSHr expression. Thus, other pathways may be involved, and the mechanism remains hypothetical. The phosphatidylinositol-3-kinase (PI3K)/AKT signaling pathway, which is known to contribute to follicle activation, may also mediate the effects of DCI [21,22]. This remains a testable hypothesis for future investigation.

MI, known as a second messenger of FSH, did not significantly affect follicle growth or E2 levels in this study. Importantly, the physiological enrichment of MI in follicular fluid does not necessarily imply that additional MI supplementation will further enhance follicle development under all in vitro conditions. One explanation is that our culture system was likely MI-sufficient at baseline (e.g., standard media components and/or endogenous MI retained within follicles), such that extra MI provided limited incremental benefit. In addition, the high, near-maximally effective FSH concentration used (~66–67 mIU/mL) may have introduced a ceiling effect, masking subtle MI-dependent modulation of FSH responsiveness or cellular metabolism. Moreover, the MI concentrations applied (20–50 μM) were selected to reflect physiological or PCOS follicular-fluid levels in vivo, whereas many in vitro maturation studies use millimolar concentrations [23]. Future studies should therefore evaluate MI in a dose–response manner under conditions with greater dynamic range (e.g., lower FSH stimulation and/or defined MI-depleted media, with careful control of osmolarity) and, to improve translational relevance, in PCOS-relevant endocrine–metabolic contexts. Taken together, the lack of an MI "add-on" effect in our MI-sufficient, high-FSH culture system should not be interpreted as evidence against MI's physiological importance in vivo, nor does it support clinical efficacy in PCOS without validation in PCOS-relevant models and human systems.

Several limitations should be emphasized. First, the single-follicle murine model has inherent constraints regarding direct clinical extrapolation: mice are polyovulatory and exhibit multifollicular ovarian morphology, whereas humans are monovulatory, and murine follicles have shorter growth periods and distinct endocrine sensitivities compared with human follicles. Second, the experimental sample size was small and multiple follicles were obtained per animal; while this approach aligns with the 3Rs principle, it may limit power and raises the possibility of non-independence among follicles derived from the same mouse. Third, the high, maximally effective FSH concentration used to support robust growth may have reduced sensitivity to detect subtle MI-dependent effects or smaller treatment differences. Fourth, only two concentrations of each inositol were tested, and MI:DCI combination ratios, long-term oocyte competence endpoints, or downstream signaling readouts were not evaluated. Finally, mechanistic interpretation remains limited because PI3K/AKT (or other pathway) activation was not directly measured; these pathways should be interrogated experimentally in future work.

In conclusion, our single-follicle culture experiments provide in vitro evidence for a modest, stage-dependent effect of DCI on early folliculogenesis, characterized by facilitation of the secondary-to-early antral transition, while MI showed no detectable effect under the present high-FSH conditions. These findings are mechanistic and should not be extrapolated to clinical efficacy in PCOS without validation in PCOS-relevant animal models and human follicle systems. Additional research is required to clarify the detailed mechanisms through which inositols regulate follicular development.

## Acknowledgments

We would like to thank Editage (www.editage.jp) for English language editing.

## Author contributions

**Conceptualization:** Aika Korai, Tsuyoshi Baba, Fukiko Kasuga-Yamashita, Sachiko Nagao, Yuya Fujibe, Miyuki Morishita, Yoshika Kuno, Tasuku Mariya, Keiko Ikeda, Hiroyuki Honnma, Toshiaki Endo, Tsuyoshi Saito.

**Data curation:** Aika Korai, Tsuyoshi Baba, Fukiko Kasuga-Yamashita, Sachiko Nagao, Yuya Fujibe, Miyuki Morishita, Yoshika Kuno, Tasuku Mariya, Keiko Ikeda, Hiroyuki Honnma, Toshiaki Endo, Tsuyoshi Saito.

**Formal analysis:** Aika Korai, Tsuyoshi Baba, Fukiko Kasuga-Yamashita, Sachiko Nagao, Yuya Fujibe, Miyuki Morishita, Yoshika Kuno, Tasuku Mariya, Keiko Ikeda, Hiroyuki Honnma, Toshiaki Endo, Tsuyoshi Saito.

**Funding acquisition:** Aika Korai, Tsuyoshi Baba, Fukiko Kasuga-Yamashita, Sachiko Nagao, Yuya Fujibe, Miyuki Morishita, Yoshika Kuno, Tasuku Mariya, Keiko Ikeda, Hiroyuki Honnma, Toshiaki Endo, Tsuyoshi Saito.

**Investigation:** Aika Korai, Tsuyoshi Baba, Fukiko Kasuga-Yamashita, Sachiko Nagao, Yuya Fujibe, Miyuki Morishita, Yoshika Kuno, Tasuku Mariya, Keiko Ikeda, Hiroyuki Honnma, Toshiaki Endo, Tsuyoshi Saito.

**Methodology:** Aika Korai, Tsuyoshi Baba.

**Project administration:** Aika Korai, Tsuyoshi Baba, Fukiko Kasuga-Yamashita, Sachiko Nagao, Yuya Fujibe, Miyuki Morishita, Yoshika Kuno, Tasuku Mariya, Keiko Ikeda, Hiroyuki Honnma, Toshiaki Endo, Tsuyoshi Saito.

**Resources:** Aika Korai, Tsuyoshi Baba, Fukiko Kasuga-Yamashita, Sachiko Nagao, Yuya Fujibe, Miyuki Morishita, Yoshika Kuno, Tasuku Mariya, Keiko Ikeda, Hiroyuki Honnma, Toshiaki Endo, Tsuyoshi Saito.

**Software:** Aika Korai, Tsuyoshi Baba, Fukiko Kasuga-Yamashita, Sachiko Nagao, Yuya Fujibe, Miyuki Morishita, Yoshika Kuno, Tasuku Mariya, Keiko Ikeda, Hiroyuki Honnma, Toshiaki Endo, Tsuyoshi Saito.

**Validation:** Aika Korai, Tsuyoshi Baba.

**Visualization:** Aika Korai, Tsuyoshi Baba.

**Writing – original draft:** Aika Korai, Tsuyoshi Baba.

**Writing – review & editing:** Aika Korai, Tsuyoshi Baba.

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
