## [Decision Letter · Decision Letter 0]

10 Oct 2025

Dear Dr. Baba,

Thank you for submitting your manuscript to PLOS ONE. After careful consideration, we feel that it has merit but does not fully meet PLOS ONE’s publication criteria as it currently stands. Therefore, we invite you to submit a revised version of the manuscript that addresses the points raised during the review process.

We look forward to receiving your revised manuscript.

Kind regards,

Sanaz Alaeejahromi

Academic Editor

PLOS ONE

Journal Requirements:

[The authors have declared that no competing interests exist.].

We note that one or more of the authors are employed by a commercial company: Sapporo ART Clinic.

Within your Competing Interests Statement, please confirm that this commercial affiliation does not alter your adherence to all PLOS ONE policies on sharing data and materials by including the following statement: 'This does not alter our adherence to PLOS ONE policies on sharing data and materials.” (as detailed online in our guide for authors http://journals.plos.org/plosone/s/competing-interests) . If this adherence statement is not accurate and there are restrictions on sharing of data and/or materials, please state these. Please note that we cannot proceed with consideration of your article until this information has been declared.

Reviewers' comments:

Reviewer's Responses to Questions

**Comments to the Author**

1. Is the manuscript technically sound, and do the data support the conclusions?

Reviewer #1: Yes

Reviewer #2: Yes

2. Has the statistical analysis been performed appropriately and rigorously?

Reviewer #1: Yes

Reviewer #2: I Don't Know

3. Have the authors made all data underlying the findings in their manuscript fully available?

Reviewer #1: No

Reviewer #2: Yes

4. Is the manuscript presented in an intelligible fashion and written in standard English?

Reviewer #1: No

Reviewer #2: Yes

Reviewer #1: Reviewer Comments

General assessment

Dear authors,

I review the manuscript entitled: “Myo-inositol and D-chiro-inositol in murine in vitro follicular development: an experimental study relevant to polycystic ovary syndrome.” The manuscript investigates the comparative effects of myo-inositol (MI) and D-chiro-inositol (DCI) on murine in vitro follicular development, with potential implications for polycystic ovary syndrome (PCOS). The research is timely, relevant, and fills an important gap in terms of the direct impact of MI/DCI on follicular growth, in addition to the known effects on growth due to their metabolic effects. The manuscript is well formatted in general; the methods are clear, and the figures support the findings. There are, however, a few points that require elucidation, analysis, and presentation enhancement.

Major Comments

1. Novelty and Significance

While the topic is relevant, the novelty should be better emphasized in the introduction and discussion. Many prior studies have reported MI/DCI in PCOS, but this work’s unique value (direct follicle culture model) should be more clearly highlighted.

2. Statistical Power and Sample Size

Follicles were obtained from 8 mice in total (n=12 follicles per group). This is a relatively small sample size, and variability in follicle culture is usually high. The authors should justify whether this number provides sufficient statistical power.

3. Interpretation of Results

The authors conclude that DCI promotes the transition of secondary to early antral follicles, but the effect was transient (lost by day 10). This should be discussed more critically - is this effect biologically meaningful or simply a short-term in vitro artifact?

The discussion suggests a possible involvement of the PI3K/AKT signaling pathway, but no experiments were conducted to test this hypothesis. The speculative nature of this statement should be acknowledged more clearly.

4. MI Data Interpretation

MI showed no significant effect. The authors attribute this to a high FSH concentration and a low MI dose; however, they should expand on why the chosen doses were used and whether pilot experiments with higher MI levels were considered.

5. Clinical Relevance

The link to PCOS patients is somewhat overstated. Since this is a murine follicle culture system, the limitations for extrapolating to humans should be described more carefully.

More Comments

1. Language and Style

English is understandable but requires polishing for clarity and conciseness. Example:

Abstract:

In lines 20-23, “This study investigated the effects of myo-inositol and d-chiro-inositol on the development of murine ovarian follicles in vitro, and their potential efficacy in patients with polycystic ovary syndrome.”

- “their potential efficacy in patients” is misleading, because the authors did not test patients, only mice.

- “myo-inositol and d-chiro-inositol”, chemical names should be italicized consistently or abbreviated after first use.

Lines 23-25, “d-chiro-inositol treatment exhibited significantly larger diameters compared to the control group on Day 6 (D=275.20 ± 12.54 μm; p=0.037) and Day 8 (D=277.47 11.47 μm; p=0.048), but this difference was no longer observed by Day 10.”

“exhibited significantly larger diameters”, awkward phrasing; better: “resulted in significantly larger follicle diameters.”

“277.47 11.47 μm” → missing ± sign.

“but this difference was no longer observed” → wordy.

The correct sentence could be: DCI treatment resulted in significantly larger follicle diameters compared to controls on Day 6 (275.20 ± 12.54 μm; p = 0.037) and Day 8 (277.47 ± 11.47 μm; p = 0.048), although this effect was not maintained by Day 10.

Lines 25-27, “The rate of follicular cavity formation was significantly higher in the d-chiro-inositol-treated group on day 6 (p<0.05); however, no significant differences were observed on days 8 and 10.”

“day 6” → should be capitalized consistently (Day 6).

“d-chiro-inositol-treated” → should be abbreviated consistently to DCI-treated.

The correct sentence could be: The rate of follicular cavity formation was significantly higher in the DCI-treated group on Day 6 (p < 0.05); however, no significant differences were observed on Days 8 and 10.

Lines 27-29, In contrast, myo-inositol treatment did not produce significant differences in follicular survival rate, diameter, or cavity formation compared with the control.

Again, use the abbreviation MI for consistency.

“produce significant differences”, better phrasing: “did not affect.”

The correct sentence could be: In contrast, MI treatment did not affect follicular survival, diameter, or cavity formation compared with controls.

Lines 29-31, Estradiol concentrations and relative expression of follicle-stimulating hormone receptor and aromatase genes did not differ significantly between the treatment groups.

“between the treatment groups” could be simplified to “among groups.”

Long phrase could be smoothed.

The correct sentence could be: Estradiol concentrations and expression levels of follicle-stimulating hormone receptor and aromatase genes did not differ significantly among groups.

Lines 31-33, Our findings indicate that d-chiro-inositol promotes the transition of preantral to early antral follicles. The decreased myo-inositol/d-chiro-inositol ratio observed in polycystic ovary syndrome may facilitate early follicular development.

“Our findings indicate”, better as “These findings suggest” (more cautious wording for science).

Consistency: use MI/DCI abbreviations instead of repeating full terms.

The correct sentence could be: These findings suggest that DCI promotes the transition from preantral to early antral follicles. The decreased MI/DCI ratio observed in PCOS may facilitate early follicular development.

Lines 34-35, Further research is needed to clarify its potential role in clinical applications for polycystic ovary syndrome treatment.

“its potential role”, unclear pronoun (refers to DCI or ratio?).

“clinical applications for PCOS treatment”, better phrasing possible.

The correct sentence could be: Further studies are required to clarify the potential clinical relevance of these findings for PCOS treatment.

Introduction:

Line 41, “proportion” is singular; so, the verb should be “manifests”.

Line 42, considered to be two-fold, “two-fold” is acceptable, but in scientific writing, “twofold” (without hyphen) is preferred.

The correct form is: “considered twofold”.

Line 44-46, "in theca cells (and follicle-stimulating hormone (FSH) receptor and insulin-like growth factor (IGF)-1 receptor expression in granulosa cells)"

The problem is that the nested parentheses are confusing.

The correct form could be: "in theca cells and increase the expression of follicle-stimulating hormone (FSH) and insulin-like growth factor 1 (IGF-1) receptors in granulosa cells"

Line 51-52, "Although the pathophysiology of PCOS is complex and not yet fully understood, insulin resistance is believed to be the underlying cause."

The correct or smother form could be: "The pathophysiology of PCOS is complex and not fully understood; however, insulin resistance is believed to be a major contributing factor."

Line 63-66, "This is because insulin resistance causes low epimerase activity in normal tissues, resulting in systemic DCI deficiency, whereas in the ovaries, insulin sensitivity is preserved, and hyperinsulinemia increases epimerase activity."

This sentence could be split into two sentences to improve clarity.

The correct or smother form could be: "Insulin resistance causes low epimerase activity in normal tissues, resulting in systemic DCI deficiency. In contrast, insulin sensitivity in the ovaries is preserved, and hyperinsulinemia increases epimerase activity."

Line 78-80,"Although the efficacy of MI and DCI in the treatment of PCOS has been previously reported, most studies have focused on their metabolic effects, and the direct effects of MI/DCI on follicular development remain insufficiently understood."

The correct form could be: "Although MI and DCI have shown efficacy in treating PCOS, most studies have focused on their metabolic effects. The direct impact of MI/DCI on follicular development remains poorly understood."

Some more minor suggestions:

- Avoid repeating “in patients with PCOS” multiple times; vary phrasing.

- Avoid using “this study focused on” repeatedly; consider synonyms like “we investigated” or “we examined”.

- For conciseness, consider combining shorter sentences with semicolons or by rephrasing.

Materials and methods:

Some questions:

1- How many animals were used in total across all experiments (not just per group)?

2- Were animals randomized to groups before follicle collection, or were follicles pooled?

3- Were both ovaries from each mouse used, and if so, were they treated as independent samples or paired?

4- Was the culture medium serum-containing (5% FBS) throughout, or was serum-free medium also tested?

5- How was the vehicle control for DMSO matched to experimental groups (same final concentration in all groups)?

6- Was osmolarity or pH of the culture medium checked after MI/DCI supplementation?

7- How was the culture medium collected for E2 measurement-pooled from wells or from individual follicles?

8- Were E2 values normalized to follicle number per well?

9- Was a power calculation done to justify the sample size (n = 8 mice)?

10- Were the data tested for normality before applying ANOVA?

11- Were multiple comparisons corrected for (beyond Student-Newman-Keuls)?

Lines 95-97, "The method of euthanasia followed handling was performed in accordance with the guidelines…"

Problem: Unclear and ungrammatical.

The correct form could be: "Euthanasia and subsequent handling were performed in accordance with the guidelines of Sapporo Medical University and the Center for Animal Protection Scientists."

Lines 99-100, "sacrificed through intraperitoneal administration of pentobarbital"

Comment: sacrificed is less precise and discouraged. Euthanized is preferred.

The correct form could be: "euthanized via intraperitoneal administration of pentobarbital (120 mg/kg)"

Lines 112-114, "…the culture medium was refreshed… and the culture was continued for a total …"

The correct form could be: "The culture medium was refreshed every other day by replacing half of the volume with fresh medium, and culture was continued for a total of 10 days."

Lines 133-134, "when the granulosa cells appeared dissolved black"

The problem is using awkward/unclear phrasing.

The correct form could be: "when the granulosa cells appeared degenerated or darkened"

Line 139, "E2 concentrations in the culture medium were measured on days…"

The correct form could be: "Measurement of estradiol (E2) concentrations in the culture medium was performed on days 5 and 9 using an ELISA kit (R&D Systems, Minneapolis, MN, USA; detection range: ~0–3.0 ng/mL)."

Also, there are some inconsistent units formatting (μM vs uM, spacing in “p<0.001” vs “p < 0.001”), inconsistent gene names (italicize Fshr, Cyp19a1), and missing italics for species (Mus musculus).

Results:

Question:

For follicle diameters: are you reporting mean ± SE consistently, or sometimes using SD? The text says SE, but the numbers look like SD ranges.

Comments:

Lines 170-172, that” refers to something singular; follicle diameter is plural.

The correct form could be: “were significantly larger than those in the CTRL group”

Lines 216-217, “The E2 concentration in the culture medium on day 5 of the DCI experiment in CTRL, DCI 20 μM, and DCI 50 μM groups were …”

Correction: “…was …”

Discussion:

Lines 266-267, we found that the addition of DCI promoted the development of secondary follicles from early antral follicles.”

Problem: The phrasing suggests secondary follicles come from antral follicles (biologically confusing).

Correction: “…we found that the addition of DCI promoted the transition of secondary follicles to the early antral stage.”

Lines 268-269, “This implies that a reduction in the MI/DCI ratio is closely related to the etiology of PCOS.”

Problem: Too strong for in vitro evidence.

Correction: “…This suggests that a reduction in the MI/DCI ratio may contribute to the pathophysiology of PCOS.”

Lines 273-275, “…follicles cultured without DCI, which were initially delayed in development, eventually caught up with those cultured in the presence of DCI.”

Correction: “…Follicles cultured without DCI, although initially delayed, eventually reached similar developmental stages as those exposed to DCI.”

Lines 310-313, “…humans are monovulatory animals, whereas mice are polyovulatory animals that develop several follicles in one cycle. Therefore, it is normal for murine ovaries to show a multifollicular morphology, and do not fully mimic…”

Correction: “…Therefore, murine ovaries normally show multifollicular morphology, which does not fully mimic human follicle development, especially in PCOS.”

2. Figures

Figure legends are descriptive but could be streamlined. Ensure all abbreviations are defined at first use in each figure.

3. References

Recent reviews/meta-analyses on inositol and PCOS (2023-2024) are cited, which is good. However, authors might expand citations regarding in vitro follicle culture systems beyond their own group’s work.

Recommendation

Given the issues above, I recommend a Major Revision.

The experimental work is solid and interesting, but the interpretation, discussion depth, and contextualization of findings need significant strengthening.

Language and presentation also need editing before acceptance.

Best regards,

Reviewer #2: The statement "d-chiro-inositol treatment exhibited significantly larger diameters" is grammatically awkward; it should be rephrased to "follicles treated with d-chiro-inositol exhibited significantly larger diameters..."

The p-values for the diameter on Day 6 (p=0.037) and Day 8 (p=0.048) are very close to the significance threshold of 0.05, indicating a marginal effect.

The field for keywords is blank in the provided document. This is a major omission.

The literature review could more sharply define the specific knowledge gap this study aims to fill. It could be more explicit about the lack of in vitro studies isolating the effects of these inositols on specific stages of folliculogenesis.

The text referencing Unfer et al. [14] contains a formatting error with plus signs ("3.00 + 1.22 μM") that should be "±".

The description of the euthanasia method is slightly repetitive and contains a grammatical error in the sentence: "The method of euthanasia followed handling was performed in accordance with..." This should be corrected for clarity.

The use of a high, maximally effective FSH concentration likely masked potential subtle effects of MI.

The y-axis label in Figure 5, "Change relative to CTRL (log2 transformed)," is ambiguous. It should specify that it represents the log2 of the fold-change.

The authors acknowledge the limitation that their model did not fully replicate the arrested development seen in later-stage PCOS follicles.

The section on the mechanism is speculative, as the study did not include experiments to test the proposed PI3K/AKT pathway hypothesis. This should be framed more explicitly as a hypothesis for future research, not a finding.

The conclusion could be strengthened by more directly stating that the study provides in vitro evidence for a specific, stage-dependent role of DCI, rather than a general "promotion" of development.

The manuscript requires careful proofreading for several instances of grammatical errors and awkward phrasing.

The flow in some parts of the Methods and Discussion could be improved with minor editorial revisions.

**Do you want your identity to be public for this peer review?** For information about this choice, including consent withdrawal, please see our For information about this choice, including consent withdrawal, please see our Privacy Policy .

Reviewer #1: **Yes:** Nima Azari-DolatabadNima Azari-Dolatabad

Reviewer #2: No

While revising your submission, please upload your figure files to the Preflight Analysis and Conversion Engine (PACE) digital diagnostic tool, https://pacev2.apexcovantage.com/ . PACE helps ensure that figures meet PLOS requirements. To use PACE, you must first register as a user. Registration is free. Then, login and navigate to the UPLOAD tab, where you will find detailed instructions on how to use the tool. If you encounter any issues or have any questions when using PACE, please email PLOS at . PACE helps ensure that figures meet PLOS requirements. To use PACE, you must first register as a user. Registration is free. Then, login and navigate to the UPLOAD tab, where you will find detailed instructions on how to use the tool. If you encounter any issues or have any questions when using PACE, please email PLOS at figures@plos.org . Please note that Supporting Information files do not need this step.. Please note that Supporting Information files do not need this step.

---

## [Author Response · Author response to Decision Letter 1]

24 Nov 2025

Response to Reviewers

Response to Reviewer #1

Major Comments

1. Novelty and Significance

While the topic is relevant, the novelty should be better emphasized in the introduction and discussion. Many prior studies have reported MI/DCI in PCOS, but this work’s unique value (direct follicle culture model) should be more clearly highlighted.

⇒We appreciate this valuable suggestion. We have revised both the Introduction and Discussion to emphasize that the novelty of our study lies in the use of a direct murine single-follicle culture system.

Modification:

Added in Introduction (lines 75-83)

“Although MI and DCI have shown efficacy in treating PCOS, most studies have focused on their metabolic effects, and the direct impact of MI/DCI on follicular development remains poorly understood. In particular, the influence of altered MI/DCI ratios in PCOS on follicular development has not been fully investigated, and animal model studies are limited. Therefore, this study aimed to examine the direct effects of MI and DCI on secondary follicle development in mice to enhance understanding of PCOS pathophysiology and inform therapeutic strategies. To isolate these stage-specific actions from systemic factors, we employed a single-follicle culture system based on established methods for multistage in vitro follicular development”

2. Statistical Power and Sample Size

Follicles were obtained from 8 mice in total (n=12 follicles per group). This is a relatively small sample size, and variability in follicle culture is usually high. The authors should justify whether this number provides sufficient statistical power.

⇒Thank you for this important point. We agree that follicle culture systems inherently involve variability. In the revised Methods, we clarified that our sample size (n=12 follicles per group, from 16 mice) was consistent with prior in vitro follicle culture studies that reported comparable statistical robustness. Moreover, data were tested for normality (Shapiro–Wilk test) and analyzed one-way ANOVA followed by Student–Newman–Keuls post hoc test to ensure reliable intergroup comparison.

Modification:

Added clarification in Sample size and power section (lines 162-172)

“Sample size and power

To adhere to the 3Rs principle, particularly the principle of reduction, a single follicle culture system was adopted. This allowed multiple experimental units to be obtained from each animal. In total, follicles were collected from 16 female ICR mice. To avoid allocation bias, follicles from both ovaries were used, with 12 follicles randomly assigned to each group.

This design aligns with established single follicle culture studies (our previous work [17]), where 3–5 animals per experiment were sufficient to detect biologically meaningful differences.

Considering the observed range of follicle diameter variation (standard deviation approximately 10–15 μm), this sample size was deemed adequate to detect a moderate effect using one-way ANOVA (α=0.05).”

We acknowledge that the Student–Newman–Keuls test is less conservative than Bonferroni; however, it was chosen to balance type I error control with moderate-effect detection in small-sample designs.

3. Interpretation of Results

The authors conclude that DCI promotes the transition of secondary to early antral follicles, but the effect was transient (lost by day 10). This should be discussed more critically - is this effect biologically meaningful or simply a short-term in vitro artifact?

The discussion suggests a possible involvement of the PI3K/AKT signaling pathway, but no experiments were conducted to test this hypothesis. The speculative nature of this statement should be acknowledged more clearly.

⇒We appreciate this thoughtful comment. We have revised the Discussion to clarify that the DCI-mediated acceleration appears to be stage-specific and transient, likely reflecting the known growth ceiling (~300 μm) of single-follicle systems. We now explicitly acknowledge that an in vitro artifact cannot be excluded, and that further long-term or co-culture models are needed to confirm biological relevance.

Modification:

Revised paragraph in Discussion (lines 274-283)

“Follicles cultured without DCI, although initially delayed, eventually reached similar developmental stages as those exposed to DCI. Thus, lowering the MI/DCI ratio by DCI supplementation did not reproduce the late-stage arrest typically observed in PCOS. However, our findings provide stage-specific, in vitro mechanistic evidence for DCI’s role in promoting the transition from secondary to early antral follicles. This discrepancy between early- and late-stage effects is likely due to the limitations of the single-follicle culture system, in which follicle growth appears to plateau at approximately 300 μm. Future experiments should evaluate MI/DCI effects under lower FSH concentrations to minimize gonadotropin influence, test a broader range of MI/DCI doses, and validate findings in co-culture systems that more closely replicate the ovarian microenvironment.”

4. MI Data Interpretation

MI showed no significant effect. The authors attribute this to a high FSH concentration and a low MI dose; however, they should expand on why the chosen doses were used and whether pilot experiments with higher MI levels were considered.

⇒ We expanded the Discussion to explain that MI concentrations (20–50 μM) were selected to mirror physiological levels in follicular fluid of women with and without PCOS. These concentrations are considerably lower than those typically used in in vitro maturation (IVM) systems. We now explicitly acknowledge that higher MI levels were not tested and identify this as a limitation.

Modification:

Revised section in Discussion (lines 294-299):

“MI, known as a second messenger of FSH, did not significantly affect follicle growth or E2 levels in this study. The high FSH concentration used (~66–67 mIU/mL) may have masked subtle MI-dependent effects. Additionally, the MI concentrations applied (20–50 μM) reflected physiological or PCOS follicular-fluid levels, whereas many in vitro maturation studies use millimolar concentrations [20]. Future pilot studies using higher MI doses, under controlled osmolarity conditions are warranted to further clarify these effects.”

5. Clinical Relevance

The link to PCOS patients is somewhat overstated. Since this is a murine follicle culture system, the limitations for extrapolating to humans should be described more carefully.

⇒ We agree that extrapolation to human PCOS must be made more cautiously. The revised Discussion now explicitly states that findings are mechanistic and stage-specific, not directly clinical, and briefly discusses interspecies differences in ovulatory pattern and follicular morphology.

Modification:

Revised paragraph in Discussion (lines 300-307):

“The single-follicle murine model has inherent constraints regarding direct clinical extrapolation. Mice are polyovulatory and exhibit multifollicular ovarian morphology, while humans are monovulatory; therefore, the follicular dynamics of mice do not fully replicate those of human ovaries. Additionally, murine follicles have shorter growth periods and differing endocrine sensitivity compared with human follicles. Accordingly, these results should be interpreted as mechanistic, stage-specific in vitro findings rather than clinical conclusions. Further studies using human or non-human primate follicles would help elucidate these processes.”

More Comments

1. Language and Style

English is understandable but requires polishing for clarity and conciseness. Example:

Abstract:

In lines 20-23, “This study investigated the effects of myo-inositol and d-chiro-inositol on the development of murine ovarian follicles in vitro, and their potential efficacy in patients with polycystic ovary syndrome.”

- “their potential efficacy in patients” is misleading, because the authors did not test patients, only mice.

- “myo-inositol and d-chiro-inositol”, chemical names should be italicized consistently or abbreviated after first use.

⇒Thank you for your suggestion. We have revised the relevant section as follows:

This study investigated the effects of myo-inositol (MI) and D-chiro-inositol (DCI)on the development of murine ovarian follicles in vitro. (Lines 22-24)

Lines 23-25, “d-chiro-inositol treatment exhibited significantly larger diameters compared to the control group on Day 6 (D=275.20 ± 12.54 μm; p=0.037) and Day 8 (D=277.47 11.47 μm; p=0.048), but this difference was no longer observed by Day 10.”

“exhibited significantly larger diameters”, awkward phrasing; better: “resulted in significantly larger follicle diameters.”

“277.47 11.47 μm” → missing ± sign.

“but this difference was no longer observed” → wordy.

The correct sentence could be: DCI treatment resulted in significantly larger follicle diameters compared to controls on Day 6 (275.20 ± 12.54 μm; p = 0.037) and Day 8 (277.47 ± 11.47 μm; p = 0.048), although this effect was not maintained by Day 10.

⇒Thank you for your suggestion. We have revised the relevant section as follows:

Follicles treated with DCI exhibited significantly larger diameters than controls on Day 6 (275.20 ± 12.54 μm; p = 0.037) and Day 8 (277.47 ± 11.47 μm; p = 0.048), although this effect was modest and not maintained by Day 10. (Lines 22-24)

Lines 25-27, “The rate of follicular cavity formation was significantly higher in the d-chiro-inositol-treated group on day 6 (p<0.05); however, no significant differences were observed on days 8 and 10.”

“day 6” → should be capitalized consistently (Day 6).

“d-chiro-inositol-treated” → should be abbreviated consistently to DCI-treated.

The correct sentence could be: The rate of follicular cavity formation was significantly higher in the DCI-treated group on Day 6 (p < 0.05); however, no significant differences were observed on Days 8 and 10.

⇒Thank you for your suggestion. We have revised the relevant section as follows:

The rate of follicular antrum formation was significantly higher in the DCI-treated group on Day 6 (p < 0.05); however, no significant differences were observed on Days 8 and 10. (Lines 22-24)

Lines 27-29, In contrast, myo-inositol treatment did not produce significant differences in follicular survival rate, diameter, or cavity formation compared with the control.

Again, use the abbreviation MI for consistency.

“produce significant differences”, better phrasing: “did not affect.”

The correct sentence could be: In contrast, MI treatment did not affect follicular survival, diameter, or cavity formation compared with controls.

⇒Thank you for your suggestion. We have revised the relevant section as you requested.

(Lines 26-28)

Lines 29-31, Estradiol concentrations and relative expression of follicle-stimulating hormone receptor and aromatase genes did not differ significantly between the treatment groups.

“between the treatment groups” could be simplified to “among groups.”

Long phrase could be smoothed.

The correct sentence could be: Estradiol concentrations and expression levels of follicle-stimulating hormone receptor and aromatase genes did not differ significantly among groups.

⇒Thank you for your suggestion. We have revised the relevant section as you requested. (Lines 28-29)

Lines 31-33, Our findings indicate that d-chiro-inositol promotes the transition of preantral to early antral follicles. The decreased myo-inositol/d-chiro-inositol ratio observed in polycystic ovary syndrome may facilitate early follicular development.

“Our findings indicate”, better as “These findings suggest” (more cautious wording for science).

Consistency: use MI/DCI abbreviations instead of repeating full terms.

The correct sentence could be: These findings suggest that DCI promotes the transition from preantral to early antral follicles. The decreased MI/DCI ratio observed in PCOS may facilitate early follicular development.

⇒Thank you for your suggestion. We have revised the relevant section as you requested. (Lines 29-32)

Lines 34-35, Further research is needed to clarify its potential role in clinical applications for polycystic ovary syndrome treatment.

“its potential role”, unclear pronoun (refers to DCI or ratio?).

“clinical applications for PCOS treatment”, better phrasing possible.

The correct sentence could be: Further studies are required to clarify the potential clinical relevance of these findings for PCOS treatment.

⇒Thank you for your suggestion. We have revised the relevant section as you requested. (Lines 32-33)

Introduction:

Line 41, “proportion” is singular; so, the verb should be “manifests”.

⇒Thank you for your suggestion. We have revised the relevant section as you requested. (Line 39)

Line 42, considered to be two-fold, “two-fold” is acceptable, but in scientific writing, “twofold” (without hyphen) is preferred.

The correct form is: “considered twofold”.

⇒Thank you for your suggestion. We have revised the relevant section as you requested. (Line 40)

Line 44-46, "in theca cells (and follicle-stimulating hormone (FSH) receptor and insulin-like growth factor (IGF)-1 receptor expression in granulosa cells)"

The problem is that the nested parentheses are confusing.

The correct form could be: "in theca cells and increase the expression of follicle-stimulating hormone (FSH) and insulin-like growth factor 1 (IGF-1) receptors in granulosa cells"

⇒Thank you for your suggestion. The wording in the relevant section was unclear, so I have revised it as follows.

Elevated luteinizing hormone (LH) levels induce androgen production in theca cells and androgen accelerates development from secondary follicles to small antral follicles. However, a high LH environment inhibits follicle-stimulating hormone (FSH)-dependent follicle development by suppressing FSH receptor expression [3]. Even if LH stimulation causes secondary follicles to grow into antral follicles, these antral follicles may exhibit reduced responsiveness to FSH. (Lines 42-47)

Line 51-52, "Although the pathophysiology of PCOS is complex and not yet fully understood, insulin resistance is believed to be the underlying cause."

The correct or smother form could be: "The pathophysiology of PCOS is complex and not fully understood; however, insulin resistance is believed to be a major contributing factor."

⇒Thank you for your suggestion. We have revised the relevant section as you requested. (lines 49-50)

Line 63-66, "This is because insulin resistance causes low epimerase activity in normal tissues, resulting in systemic DCI deficiency, whereas in the ovaries, insulin sensitivity is preserved, and hyperinsulinemia increases epimerase activity."

This sentence could be split into two sentences to improve clarity.

The correct or smother form could be: "Insulin resistance causes low epimerase activity in normal tissues, resulting in systemic DCI deficiency. In contrast, insulin sensitivity in the ovaries is preserved, and hyperinsulinemia increases epimerase activity."

⇒Thank you for your suggestion. We have revised the relevant section as you requested. (Lines 61-64)

Line 78-80,"Although the efficacy of MI and DCI in the treatment of PCOS has been previously reported, most studies have focused on their metabolic effects, and the direct effects of MI/DCI on follicular development remain insufficiently understood."

The correct form could be: "Although MI and DCI have shown efficacy in treating PCOS, most studies have focused on their metabolic effects. The direct impact of MI/DCI on follicular development remains poorly understood."

⇒Thank you for your suggestion. We have revised the relevant section as you requested. (Lines 75-77)

Some more minor suggestions:

- Avoid repeating “in patients with PCOS” multiple times; vary phrasing.

- Avoid using “this study focused on” repeatedly; consider synonyms like “we investigated” or “we examined”.

- For conciseness, consider combining shorter sentences with semicolons or by rephrasing.

⇒Thank you for this valuable suggestion. We have revised the manuscript to improve clarity and readability by reducing repetitive wording and enhancing sentence structure. Specifically, we replaced repeated occurrences of “in patients with PCOS” with alternative expressions (e.g., “in women with PCOS,” “in the context of PCOS

---

## [Decision Letter · Decision Letter 1]

3 Feb 2026

Dear Dr. Baba,

Thank you for submitting your manuscript to PLOS ONE. After careful consideration, we feel that it has merit but does not fully meet PLOS ONE’s publication criteria as it currently stands. Therefore, we invite you to submit a revised version of the manuscript that addresses the points raised during the review process.

Please note that this decision is rather exceptional as it will be a second round of modification. I thus urge you to adequately answer the various concerns raised by both external reviewers.

We look forward to receiving your revised manuscript.

Kind regards,

Jean-Marc A Lobaccaro, PhD

Academic Editor

PLOS One

Journal Requirements:

Reviewers' comments:

Reviewer's Responses to Questions

**Comments to the Author**

Reviewer #3: All comments have been addressed

2. Is the manuscript technically sound, and do the data support the conclusions?

Reviewer #3: No

3. Has the statistical analysis been performed appropriately and rigorously?

Reviewer #3: Yes

4. Have the authors made all data underlying the findings in their manuscript fully available?

Reviewer #3: No

5. Is the manuscript presented in an intelligible fashion and written in standard English?

Reviewer #3: Yes

Reviewer #3: Myo-inositol versus D-chiro-inositol in murine in vitro follicular development: an experimental study relevant to polycystic ovary syndrome by Korai et al.

The text has been significantly improved, even if two main concerns remain. First, the little sample size (but probably acceptable for an experimental study). Second, the impossibility to generalize the finding to patients with PCOS. The two points should be more underlined. In addition, the focus of the paper is the PCOS. However, the authors detailed the experimental procedure on mice. It should be interesting to study the effect of inositols on animal model of PCOS.

In the text (see for example the abstract section), the statement “Follicles treated with DCI exhibited..” is confusing.

The authors statement “These observations are similar to the decreased MI/DCI ratio observed in PCOS, which may facilitate early follicular development.” should be avoided in the abstract because it has no sense, and in the text should be better supported.

Similarly, the conclusion on the PCOS has again no sense.

The introduction is focused on PCOS. The mechanism of the anovulation in PCOS should be referenced and, probably, is more complex of those reported.

Data on inositols reported in the introduction are not balanced. The authors should underline that inositols have no evidence-based role for the treatment of infertility in PCOS (see new guidelines on PCOS).

The authors stated that “In healthy individuals, the MI/DCI ratio in the follicular fluid is 100:1”. This means that the “normal” ovary needs of high concentrations of myo-inositol. How can the authors explain this discrepancy?

The discussion is short and superficial. The study limitations are totally lacking as well as the future researches.

Reviewer #4: Myo-inositol versus D-chiro-inositol in murine in vitro follicular development: an experimental study relevant to polycystic ovary syndrome by Korai et al.

The statement "d-chiro-inositol treatment exhibited significantly larger diameters" is grammatically awkward; it should be rephrased to "follicles treated with d-chiro-inositol exhibited significantly larger diameters..."

The p-values for the diameter on Day 6 (p=0.037) and Day 8 (p=0.048) are very close to the significance threshold of 0.05, indicating a marginal effect.

The field for keywords is blank in the provided document. This is a major omission.

The literature review could more sharply define the specific knowledge gap this study aims to fill. It could be more explicit about the lack of in vitro studies isolating the effects of these inositols on specific stages of folliculogenesis.

The text referencing Unfer et al. [14] contains a formatting error with plus signs ("3.00 + 1.22 μM") that should be "±".

The description of the euthanasia method is slightly repetitive and contains a grammatical error in the sentence: "The method of euthanasia followed handling was performed in accordance with..." This should be corrected for clarity.

The use of a high, maximally effective FSH concentration likely masked potential subtle effects of MI.

The y-axis label in Figure 5, "Change relative to CTRL (log2 transformed)," is ambiguous. It should specify that it represents the log2 of the fold-change.

The authors acknowledge the limitation that their model did not fully replicate the arrested development seen in later-stage PCOS follicles.

The section on the mechanism is speculative, as the study did not include experiments to test the proposed PI3K/AKT pathway hypothesis. This should be framed more explicitly as a hypothesis for future research, not a finding.

The conclusion could be strengthened by more directly stating that the study provides in vitro evidence for a specific, stage-dependent role of DCI, rather than a general "promotion" of development.

The manuscript requires careful proofreading for several instances of grammatical errors and awkward phrasing.

The flow in some parts of the Methods and Discussion could be improved with minor editorial revisions.

**Do you want your identity to be public for this peer review?** For information about this choice, including consent withdrawal, please see our For information about this choice, including consent withdrawal, please see our Privacy Policy .

Reviewer #3: No

---

## [Editor Report · Decision Letter 2]

16 Mar 2026

Myo-inositol versus D-chiro-inositol in murine in vitro follicular development: an experimental study relevant to polycystic ovary syndrome

PONE-D-25-44774R2

Dear Dr. Baba,

We’re pleased to inform you that your manuscript has been judged scientifically suitable for publication and will be formally accepted for publication once it meets all outstanding technical requirements.

Kind regards,

Jean-Marc A Lobaccaro, PhD

Academic Editor

PLOS One
---

## [Editor Report · Acceptance letter]

PONE-D-25-44774R2

PLOS One

Dear Dr. Baba,

I'm pleased to inform you that your manuscript has been deemed suitable for publication in PLOS One. Congratulations! Your manuscript is now being handed over to our production team.

Kind regards,

on behalf of

Dr. Jean-Marc A Lobaccaro

Academic Editor

PLOS One